# Effect of Calcareous Marine Algae Buffer on High-Producing Dairy Cows during Peak Lactation

**DOI:** 10.3390/ani14060897

**Published:** 2024-03-14

**Authors:** Radko Loučka, Václav Jambor, Hana Synková, Petr Homolka, Dana Kumprechtová, Veronika Koukolová, Petra Kubelková, Alena Výborná, Yvona Tyrolová, Filip Jančík

**Affiliations:** 1Department of Animal Feeding and Nutrition, Institute of Animal Science, 104 00 Prague, Czech Republic; loucka.radko@vuzv.cz (R.L.); homolka.petr@vuzv.cz (P.H.); kumprechtova.dana@vuzv.cz (D.K.); koukolova.veronika@vuzv.cz (V.K.); kubelkova.petra@vuzv.cz (P.K.); vyborna.alena@vuzv.cz (A.V.); tyrolova.yvona@vuzv.cz (Y.T.); 2NutriVet Ltd., 691 23 Pohořelice, Czech Republic; jambor.vaclav@nutrivet.cz (V.J.); synkova.hana@nutrivet.cz (H.S.)

**Keywords:** subacute ruminal acidosis, total mixed ration, feeding, rumination

## Abstract

**Simple Summary:**

High-producing dairy cows face an increased risk of subacute ruminal acidosis, which negatively affects the general health, feed intake, and the quantity and quality of milk produced. The aim of this study was to investigate the effect of calcareous marine algae (*Lithotamium calcareum*)-based rumen content buffer included in total mixed ration, fed to 34 high-producing, peak lactation Holstein dairy cows (group A, *n* = 17; group B, *n* = 17). It was hypothesized that through a rumen content buffering effect, buffer would improve feeding behavior, feed intake, rumen function, and performance. Differences between the experimental and control total mixed ration in most parameters under study (feed intake, rumen content acidity, feces composition, milk yield, and milk composition) were statistically insignificant. However, statistically significant differences were noted in the length of eating and chewing epizodes (feeding time; rumination time) between groups. Feed intake was in line with this, although the difference was only numerical and statistically insignificant. This may simply mean that the buffer effects takes some time to show but persists after the buffer withdrawal. One of the reasons for not achieving a significant improvement in other parameters might be that the cows were fed a typical, well-balanced ration that did not induce acidosis.

**Abstract:**

The aim of the study was to investigate the effect of calcareous marine algae (*Lithotamium calcareum*)-based rumen content buffer (CMA) included in concentrated feed within total mixed ration (TMR), fed to 34 peak lactation (87–144 days in milk) Holstein dairy cows, randomized into two groups (group A, *n* = 17; group B, *n* = 17), wearing collars with accelerometers, and housed a in barn with automatic feed-weigh troughs. During the first phase P1, group A received TMR with CMA (TMR-E) and group B was fed TMR without the buffer (TMR-C). For P2, the treatments in the groups were exchanged. Feed intake, feeding time (FT), rumination time (RT), milk yield, milk composition, and rumen pH were measured by barn technologies, and rumen fluid and feces composition were analyzed in the laboratory. Differences between the TMR-E and TMR-C in most parameters under study were statistically insignificant, except overall FT and RT, which differed significantly between the groups. Group A, feeding at P1 by TMR-E, exhibited higher FT and RT than Group B (202 min/cow/day vs. 184 min/cow/day, and 486 min/cow/day vs. 428 min/cow/day, respectively). The RT significantly increased after switching from TMR-C to TMR-E. This implies that the buffer effect is delayed and persists after the withdrawal. In the group of cows that received control TMR without buffer in the first phase, RT and milk protein content increased significantly in the first week after the addition of buffer.

## 1. Introduction

During peak lactation, the rumen environment in high-yielding dairy cows is variable. There is an increased risk of subacute ruminal acidosis (SARA), which causes significant economic losses in dairy herds. Reduced rumen pH and diurnal variation are considered to be major factors influencing the development of SARA [1].

According to a meta-analysis [2], there are multiple causes of SARA. The main ones are the cow and dietary mistakes. No single dietary factor solely drives the physiochemistry of the rumen; there is a combination of several factors [3].

Over the past decade, several studies have investigated related topics and reported rumen pH data obtained from continuous measurement with rumen pH sensors instead of single-shot samples commonly used in the past, providing new data on the duration of reduced pH periods. It is generally accepted that SARA occurs when a rumen pH < 5.6 lasts longer than 3 h/day [4,5]. There are some other SARA definitions, e.g., according to Valente et al. [1], the risk threshold for SARA is pH below 5.8. In another study [6], the threshold for SARA is pH below 5.8 (measured directly in the rumen), lasting more than 5.2 h/day. To date, no official pH threshold for SARA has been established in dairy cows for two main reasons: the imprecision of measurement techniques and high variability of rumen pH among cows. There have been several meta-analyses and reviews on SARA [7,8,9,10,11,12,13].

Some studies [14,15,16,17] already investigated the effects of rumen buffers on pH and SARA prevention. Loučka et al. [18] compared the inclusion of buffers and straw in TMR. Not all the studies have shown positive effects of buffers. Several studies have shown that there is large inter-cow variability in the intensity of chewing and the nature of buffering, with some cows being sensitive to SARA and others tolerant to it [19,20]. In a ruminant herd, there are often large differences in SARA severity among animals, even when fed the same diet [21,22].

An important part of SARA prevention is the measurement of performance indicators [23,24]; feeding behavior [25,26,27]; milk quality [28,29]; ruminal fermentation [30]; and, where appropriate, residual nutrient levels in feces.

An indirect method to diagnose SARA is the monitoring of chewing movements with an accelerometer placed in the neck collar. In addition, some indirect practical indicators can be used such as the appearance of feces and the milk fat/protein ratio. [31].

The hypothesis was that when buffer is added to a concentrated mixture in TMR of dairy cows, this would affect milk production and quality, feed consumption, eating, and rumination time and other parameters such as rumen acidity or fecal composition.

The aim of the study was to investigate the effect of calcareous marine algae (*Lithotamium calcareum*) buffer inclusion in total mixed ration (TMR) on the feeding behavior and performance of high-yielding Holstein dairy cows.

## 2. Materials and Methods

The design of this study was similar to the one published by Loučka at al. [18]. The trial was carried out in the same barn with the same technology but included different dairy cows and different TMR, and its objective was different, too.

### 2.1. Technical Details

The experiment was carried out on the Experimental Farm of Institute of Animal Science in Prague (50°05″ N and 14°27″ E; altitude 287 m above sea level; 8.4 °C daily mean; and 526 mm average precipitation). The cows were housed in an experimental barn equipped with the roughage intake control system (RIC) (RIC2DISCOVER, Hokofarm Group, Insentec, Marknesse, The Netherlands). The RIC system consists of a feed-weigh trough (tensometric feed trough, TFT) combined with an intelligent feeding gate to gather individual cow feed intake. During the trial, the RIC system continuously monitored and recorded the feed intake, duration of eating, and number and time of visits to the feed trough, for each cow separately. TMR was placed into the TFTs six times per day. All the experimental cows had an unrestricted access to feed and water.

All the cows were monitored daily for health indicators, especially for clinical signs of acidosis, both visually and using the above precision livestock farming (PLF) methods. The dairy cows in the barn had collars with accelerometric sensors, a PLF technology which continuously records parameters such as feed ingestion (feeding time, FT) and cud chewing (rumination time, RT) via the Vitalimeter device (AGROSOFT, Tábor, Czech Republic).

Milk yield (MY) and composition were recorded twice a day by the herd management system AfiFarm 5.5 (Afimilk Ltd., Kibbutz Afikim, Israel) for each cow. MY was measured by an electronic lactometer module (Afimilk), and fat and protein content were measured by the milk analysis module (Afilab).

### 2.2. Dairy Cows

The experimental protocol was approved by the institutional Animal Care and Use Committee, Institute of Animal Science, Prague, Czech Republic (Act No. 359/2012 Coll.). Thirty-four high-yielding Holstein dairy cows were included in the experiment. During the 2-week adaptation period, the cows were getting used to the barn environment and TFT technology. The two experimental phases (P1, P2) followed, each lasting 3 weeks. The cows were randomized into two groups (A, B) by the pairing method according to parity, days in milk (DIM), daily milk yield, and live weight, so that each cow pair was had similar values at the beginning of the first experimental phase (Table 1). Of each group of 17 animals, 4 were primiparous.

MY averaged more than 40 liters at the start of the trial. The initial average milk fat and protein content was 2.6% and 2.8%, respectively. All the automated measurements were performed daily in all the cows. The number of cows (34) in the experimental barn did not exceed its capacity. Rumen fluid, milk and feces samples for laboratory analyses were taken from 12 cows.

### 2.3. Feeding

During the 14-day adaptation period, all the cows were fed the control diet (TMR-C) that was formulated by the feed formulation software based on NRC [32].

In the first trial phase (P1), group A received the control TMR supplemented with the CMA buffer (TMR-E) and group B was fed TMR-C (without the buffer). In the second trial phase (P2), group A received TMR-C and group B received TMR-E. The CMA buffer was added into concentrated feed mixture (CFM) included in the total mixed ration. The ingredients of TMR-C and TMR-E and chemical analysis results are listed in Table 2. The CFM ingredients and analytical composition are given in Table 3. The CMA buffer contains a blend of calcareous marine algae (*Lithotamium calcareum*), *Yucca Schidigera* powder, cristobalite, fulvic acid (from the drinking water industry), calcium propionate (E282) 0.00025% as preservative, a blend of aromatic ingredients 0.01%, and anti-caking agent Silica (E551a) 0.35%. Wheat flour was used as a carrier. The inclusion level of CMA in CFM-CMA was 2.3%. With 8.5 kg of CFM/cow/day, this makes for a CMA buffer dose of 196 g/cow/day.

Dosing of TMR components was controlled by the mixer wagon Husky DS 90 software. Maize silage was made without a silage additive; alfalfa was treated by 1 g/t Formasil^®^ Alfa (*Lactobacillus plantarum* (CNCM MA 18/5U) 1.5 × 10^11^ cfu/g, *Pediococcus pentosaceus* (NCIMB 12455) 1.5 × 10^11^ cfu/g, enzymes: beta-glucanase with activity >150,000 nkat/g, and xylanase with activity >136,000 nkat/g).

TMR samples were collected during the last week of each trial phase. The chemical composition of TMR was analyzed 3 times. Fresh samples were dried for 24 h at 50 ± 2 °C and subsequently milled to pass through a 1-mm sieve. Dry matter (#934.01), ash (#942.05), crude protein (#976.05), starch (#920.40), neutral detergent fiber (#2002.04), and acid detergent fiber (#973.18) were determined according to the methods of the Association of Official Analytical Chemists [33], and crude fiber contents were determined according to the Weende’s gravimetric method [34]. Energy (MJ NEL/kg DM) was calculated based on measured chemical composition and nutrient digestibility values using the equations of Vencl et al. [35].

### 2.4. Other Parameters under Study

The samples of milk, rumen fluid, and feces were collected once during the last week of each trial phase (P1, P2) from 7 cows per group.

Milk quality was analyzed in an accredited (ČSN EN ISO/IEC 17025:2005, 2018) laboratory (MILCOM a.s. Prague, Czech Republic). Infrared spectrophotometry (ČSN 57 0530, 2010; ČSN 57 0536, 1999) was used to measure fat, protein, lactose, and non-fat milk solids, and casein, urea, and free fatty acids (FFA) were measured by the indirect MIR-FT method [28].

Rumen fluid was taken with a stomach tube (length 240 cm; diameter 2.5 cm; insertion depth 180 cm) four hours after morning feeding. The cows were restrained in a special fixation cage to ensure a safe and easy procedure. Each time 250 mL of rumen fluid were harvested, and 1 mL of toluene was added for preservation. Then, the samples were transported to laboratory where pH was measured, and rumen fluid was centrifuged at 1200 rpm for 5 min. Supernatant was transferred into a PE bottle and frozen until analysis. Rumen fermentation parameters were analyzed as follows: pH potentiometrically, using inoLab level 1 (INOLAB, WTW, Weilheim, Germany); volatile fatty acids (mmol/L of rumen fluid) by capillary electrophoresis [36], using ITP/CZE analyzer IONOSEP 2003 (RECMAN, Ostrava-Hrabůvka, Czech Republic); and ammonia nitrogen (mg N/100 g rumen fluid) spectrophotometrically, using Biochrom Libra s22 (BIOCHROM Ltd., Cambridge, UK).

Fecal samples were collected from the rectum in the amount of 0.8 kg per cow, analyzed for DM and starch content, and measured according to AOAC [33]. The pH value of fecal samples was measured with a laboratory pH meter inoLab level 1 (INOLAB) in a solution of 15 g feces mixed in 100 mL of distilled water.

### 2.5. Statistical Analysis

The experimental design was as follows: TMR-E or TMR-C was fed to 17 (group A) and 17 (group B) cows in two 21-day phases. Analysis of variance (ANOVA, STATISTICA, 10) with 2 × 2 × 2 factorial design was used [37].

Independent variables were TMR, group, and phase, and the dependent variables were the individual nutritional and performance parameters measured by respective sensors.

The associations for each item among factors were evaluated using a bivariate correlation analysis [37]. The probability of correlation (*p*-value) was calculated, and Pearson bivariate correlations [38] were considered significant at *p* < 0.05. The r coefficient values for correlation were interpreted according to Prion and Haerling [39]: very strong correlation (±0.91 to ±1.00); strong correlation (± 0.68 to ±0.90); moderate correlation (±0.36 to ±0.67); weak correlation (±0.21 to ±0.35); and negligible correlation (0 to ±0.20).

## 3. Results

### 3.1. Data from the Sensors

As shown in Table 4, the differences between TMR-C and TMR-E were not significant in any of the parameters measured. There was a difference between P1 and P2 in TMR intake (46.8 vs. 50.1 kg/cow/day) and feeding time (FT), with higher values in P2. In contrast, longer rumination time (RT) was observed in P1 than in P2 (492 vs. 421 min/cow/day). In P1, Group A fed TMR-E showed lower protein levels (2.60 vs. 2.67%) than Group B. Both FT (202 vs. 184 min/cow/day) and RT (486 vs. 428 min/cow/day) were higher in P1.

Values from sensors by Phase (Ps—start of phase, Pc—middle of phase, and Pe—end of phase) are shown in Table 5. Statistical significance (*p* < 0.05) was obtained for milk protein, but the differences between the values at the start and end of each period, whether evaluated by group or TMR, were not statistically significant. Significant differences only became apparent when assessing the transition from one TMR to another. In Group A, after switching from TMR-E to TMR-C, a significant increase in milk protein from 2.60% to 2.69% was only evident in the second week of P2, whereas the opposite was true for Group B, with a significant decrease in milk protein from 2.69% to 2.60%. Both transitions were gradual and tended to follow the original trend, i.e., after the inclusion of the buffer at the beginning of the experiment, the protein content gradually increased, although the buffer was discontinued after 21 days, or the decrease in protein content continued after the buffer was included in the second half of the experiment. In Table 5, it can be seen that in group B, which started the experiment only with the control TMR without buffer, after switching to TMR with buffer, the rumination time increased significantly.

In the first trial phase (P1), TMR-C was fed to group B, and in the second phase, (P2) was fed to group A. During P1, group A was fed TMR-E containing concentrated feed mixture (CFM) with CMA buffer, and during P2, TMR-E was fed to group B. Statistical significance (*p* < 0.05) was also obtained for RT. Differences between values at the beginning and end of phase, whether judged by group or TMR, were not statistically significant, but there was a significant difference between the initial value of Group A and Group B at P1. Group A cows had RT of 500 min, whereas Group B cows had RT of only 464 min/cow/day.

In Group A, fed TMR-E with buffer, RT decreased gradually in P1, but once the buffer was removed (in P2), RT increased from 447 to 531 and remained stable until the end of P2 (526). In group B, the transition from P1 to P2, or from TMR-C to TMR-E, did not influence RT (419 vs. 412 min/cow/day).

Correlation coefficients between the values measured by the sensors in the milking parlor; tensometric feed troughs (TFT) and neck collars (NC) are given in Table 6. Feeding time showed the highest positive correlation, although only on a moderate level [39], with time spent in TFT (r = 0.633) and number of visits to TFT (r = 0.397). Logically, MY increased with the amount of TMR consumed (r = 0.451).

### 3.2. Milk Quality Analyzed in the Laboratory

Milk quality (Table 7) was analyzed for all the dairy cows included in the experiment. The differences between TMR-C and TMR-E, and between P1 and P2, were not statistically significant. Significant differences were found only between Group A and Group B in fat-free dry matter (FFDM) and free-fatty acids (FFA) content. Both FFDM and FFA values were lower in group A than in group B.

### 3.3. Rumen Fermentation and Starch Digestion

Rumen fluid values measured in selected dairy cows (*n* = 12) are shown in Table 8. The differences between TMR-E and TMR-C for the other parameters under study were not statistically significant. There was a significant difference (*p* = 0.039) between Group A and Group B for valeric acid only (0.63 vs. 0.57 mol%). Between P1 and P2, there were significant differences, too. pH values were higher in P1 (6.57 vs. 6.0), whereas total volatile acids were higher in P2 (122.7 vs. 141.7 mM). In P2, rumen fluid contained higher levels of lactic and butyric acids and lower levels of acetic acid.

Values of pH, dry matter, and starch content of feces are given in Table 9. No differences were found either between TMRs or between the groups of cows. Differences were found only between the phases. The pH values were higher in P1 than in P2 (6.67 vs. 6.50), and the starch content was also higher in P1 (35.3 vs. 32.3 g/kg dry matter).

## 4. Discussion

The aim of the study was to determine whether the inclusion of calcareous marine algae (*Lithotamium calcareum*) in TMR would affect the performance of high-yielding Holstein dairy cows. For this purpose, the cows were randomized into two groups (A, B) at the end of adaptation phase by the pairing method according to parity, days in milk (DIM), daily milk yield, and live weight (Table 1). As shown in Table 1, the initial conditions were the same for both groups. The switch of TMR between the groups in the middle of trial ensured that the buffer supplemented ration was fed to all the animals included in the trial. The switch of TMR between the groups allowed for the evaluation of the transition from TMR with buffer to TMR without buffer and vice versa. The 21-day phases were not long enough to affect lactation curve. Another reason might have been the timing of trial during the peak lactation (DIM 87 to 144).

Using the feed formulation software based on NRC [32], TMR was formulated from raw materials commonly used for feeding high-producing dairy cows in the Czech Republic. The composition of TMR is presented in Table 2, and CFM composition is presented in Table 3. The components and nutrient contents of TMR and CFM indicated some risk of SARA because TMR was formulated for MY of 41.6 kg per day.

Estimated daily intake of fresh and dry TMR matter was 48.6 kg and 22.4 kg, respectively. The cows were fed well-balanced TMR ad libitum. Average daily milk yield for the whole trial (42 days) was 39.0 ± 5.0 kg, and average daily TMR intake was 48.5 ± 6.6 kg per cow. In terms of actual results obtained compared to those predicted (calculated), the TMR formulation was accurate because the results obtained were comparable to those predicted. Indeed, if a lactation curve persistency of 94% [40] had been taken into account in the TMR calculations, average daily milk yield of 39.0 kg would have been even more in line with the predicted one (39.1 kg). Diets with an appropriate concentrate-to-forage ratio (C:F) can provide balanced nutrition for ruminants, which leads to better feed efficiency and milk production, and well-balanced rumen microflora [41].

Whether the addition of buffer to TMR will be effective also depends on other factors, especially the individuality of animals and their response to the stimuli at a given time. Continuous measurement of performance and health indicators by methods enabling to take prompt actions may influence results. An important reason for the buffer not affecting the measured parameters in this study could be the feeding of TMR that did not induce SARA.

The experimental TMR was well formulated even in terms of C:F. The correctness of the TMR formulation is also supported by the rumen fluid values (Table 9). On average, rumen fluid pH values were 6.29 ± 0.44, which is not even close to SARA range (pH less than 5.8).

As shown in Table 4, Group A showed significantly longer overall feeding time (FT) and rumination time (RT) than Group B. In P1, the difference between TMR-E and TMR-C was not significant. However, in P2, Group A showed a significantly higher FT and RT than Group B. Overall feed intake results were in line with this, although the difference between Group A and B (49.1 kg/cow/day vs. 47.8 kg/cow/day, respectively) was not statistically significant. This might imply that it takes some time before the effect of rumen buffer shows, but after the buffer is withdrawn, its effect persists for some time.

In the other measured parameters, the differences between TMR-C and TMR-E were not significant. There were differences between the cow groups in milk protein content, FT and RT. In order to evaluate performance and physical activity indicators into a greater detail, they were also assessed in each week of each phase (Table 5). The inclusion or withdrawal of the dietary buffer had a significant impact only on milk protein and RT. Milk protein content increased when buffer was included, similarly to the findings of Neville et al. [15]. A milk quality study of Kara [29] showed that dairy cows with lower milk urea nitrogen (MUN) and milk fatty acid (MFA) content are more likely to develop SARA.

RT decreased after the buffer inclusion in Group A (from 500 to 447 min/cow/day) and increased immediately after the buffer withdrawal (531 min/cow/day). However, the differences were not significant.

Group B was fed the control TMR in the first phase; RT decreased from 412 to 350 min/day; and soon after the switching to the buffered TMR, RT increased significantly to 464 min/day. The decrease in RT after the buffer inclusion, as well as the increase in RT after the buffer withdrawal, is consistent with the study by Zhang et al. [20]. The plethora of literature published in recent years provides new insights into eating and ruminating activity of dairy cows. Lactating dairy cows spend about 4.5 h/d eating (range: 2.4–8.5 h/d) and 7 h/d ruminating (range: 2.5–10.5 h/d), with a maximum total chewing time of 16 h/d.

However, the net effect of changing chewing time on rumen buffering is likely to be rather small according to Beauchemin [10]; therefore, acidosis prevention strategies should be comprehensive.

Correlation coefficients between the values measured by sensors in milking parlor, TFT, and NC are summarized in Table 6. The highest positive correlation (r = 0.63) was found between FT (TFT) and FT (NC), although only at a moderate level [39]. These values are higher than those obtained in our previous experiment [18], where FT (TFT) was correlated with FT (NC) at r = 0.27. Logically, MY increased with the amount of TMR consumed (r = 0.451), and the correlation coefficient (r = 0.55) was higher in our previous experiment [18]. Stone et al. [42] reported a weak correlation (r = 0.22) between RT and daily MY.

According to a meta-analysis by Souza et al. [11], MY averaged 34.3 kg/d (range: 14.2–52.1) and milk fat averaged 3.47% (range: 2.20–4.60). Average dry matter intake was 23.1 kg/d (range: 15.3–32.6). Mean rumen pH was 6.1 (range: 5.3–7.0) of the 292 reported observations. These are the baseline data for comparison with our inputs and results. The meta-analysis by Souza et al. [11] suggested that RT was quadratically increased with increasing milk fat yield (partial r = 0.27) and milk fat percent (partial r = 0.17). Rumination was also increased with increasing MY, DM intake, and rumen pH, and it was quadratically related to dietary NDF and total-tract NDF digestibility (partial r = 0.10–0.27). Similar relationships were observed for rumination per unit of dry matter and NDF intake.

Milk fat yield was the highest at RT of 494 min/day, and there was no additional benefit of higher RT. In multivariate analysis, a set of variables explained 37% of RT. Overall, RT was mostly associated with milk fat. However, the correlation was only moderate [39], suggesting that rumination is not the only factor influencing optimal and stable ruminal fermentation, and therefore factors other than ruminal fermentation influence milk fat production.

According to Kleen et al. [7], SARA likely arises when high-energy TMR enters a rumen environment that is not adapted to this type of substrate. SARA can be caused by inaccurate estimation of dry matter intake (DMI), resulting in an incorrect forage to concentrate ratio, inadequate feed structure, or errors during manufacture of TMR. All this implies that the risk of SARA was low in our study, for SARA did not adversely affect the parameters measured by sensors in the barn, collar, and milking parlor.

In addition to the monitoring with barn, collar, and milking parlor sensors, the tendency to manifest SARA was supported by chemical measurements in the laboratory. Milk components (Table 7) were within normal limits [28], as were rumen fluid (Table 8) and feces (Table 9) collected from selected cows.

Differences in milk quality were found only between the groups and only for FFDM = fat-free dry matter and FFA = free fatty acids. Both parameters were significantly higher in Group B, probably due to variation among the cows.

On average, rumen fluid showed pH values of 6.29 ± 0.44, acetic acid 63.5 ± 1.35 mol%, propionic acid 19.6 ± 1.12 mol%, butyric acid 16.3 ± 0.62 mol%, and ammonia nitrogen 13.7 ± 1.01 mg N/100 g. The measured values are in agreement with those reported by Kitkas et al. [30]. They found that a one-point increase in ruminal pH was associated with a significant decrease in concentration of acetic (by 27.6 mmol/L, *p* < 0.001), propionic (by ca. 24.0 mmol/L, *p* < 0.001), and butyric (by 16.0 mmol/L, *p* < 0.001) acids. According to Kitkas et al. [30], pH decreases with increasing acidity; the higher the ratio of acetic to propionic acid, the higher the rumen pH or the lower the risk of acidosis. The ratio of acetic acid to propionic acid was higher in P2 than in P1, suggesting that in more advanced lactations there is a lower risk of acidosis in cows fed the same TMR.

Fecal pH values were 6.58 ± 0.16, and starch content was 33.8 ± 2.65 g/kg DM. According to Fredin et al. [43], starch content up to 30 g/kg DM can be considered optimal, while between 30 and 50 g/kg DM is still satisfactory. However, even a high starch content in feces does not necessarily mean that cows will develop SARA. This is evidenced by the experiment of Abeyta et al. [44]. During the experiment, all cows were fed the same TMR with a starch content of 26% DM, ad libitum. By infusing pure corn starch through a rumen cannula, 5 cows were transferred to an energy-dense diet by increasing the starch content of TMR. Starch infusion significantly decreased fecal pH (5.84 vs. 6.76) and increased fecal starch (by 22 to 96 g/kg DM) compared to baseline. Importantly, milk yield, milk constituents, and voluntary DMI remained unchanged post-infusion. However, corn starch infusion caused extensive hindgut fermentation, as indicated by a significant decrease in fecal pH.

The risk of SARA was not confirmed in our study by chemical analyses of milk, rumen fluid, or fecal starch. The low risk of SARA was probably due to the fact that the buffer supplementation had no significant effect on any of the measured parameters. Since the cows were fed a well-balanced total mixed ration and the differences in results between the cows fed TMR-C and TMR-E were not significant, it can be concluded that the inclusion of buffer in the diet was superfluous. However, this does not mean that the buffering capacity of CMA, based on calcareous marine algae (*Lithotamium calcareum*), was low; it just could not be demonstrated under the circumstances. In most studies, the buffer inclusion made a difference but not always in the same parameters. However, the experiments described in the literature had a different design or used different buffers at different doses from this study.

Sodium bicarbonate is the most popular rumen buffer because at about 0.8% dietary DM it effectively neutralizes rumen acidity, increases milk fat, and increases DM intake. However, as a soluble buffer it has a short rumen life and thus cannot effectively buffer ongoing rumen acid production for extended periods of time [16]. A meta-analysis by Hu and Murphy [45] reviewed 27 studies that evaluated the responses of dairy cows in early and mid-lactation when TMR was buffered with sodium bicarbonate. Significant interactions were observed, but it was found that milk production, protein percentage, and protein yield were not affected by the buffer, regardless of forage type.

The effect of calcareous marine algae (CMA) on rumen pH and milk production in mid-lactation dairy cows has been investigated by Bernard et al. [17], Cruywagen et al. [16], Neville et al. [15], and others. In these three studies, no differences in milk production and milk composition were observed. The results varied in detail.

Bernard et al. [17] included 87 g/day of CMA in TMR. Control TMR tended to increase milk protein compared with CMA-TMR. According to Cruywagen et al. [16], CMA supplementation of 90 g/day in TMR resulted in improved rumen pH and feed efficiency for milk production and composition. Neville et al. [15] tested MgO supplementation in addition to CMA in two experiments with dairy cows and compared the results with a buffer-free and sodium bicarbonate options. The experiments were designed in a way to ensure that the buffers used in TMR would have a significant effect on pH. In both experiments, the diets were formulated for dry matter intake (DMI) of 18 kg/cow/day, with 80 g of buffer added to the TMR. The forage-to-concentrate ratio was 45:55. In our trial, the DMI was higher, 22.4 kg/cow/day, and CMA was provided at 196 g/cow/day. Forage-to-concentrate ratio was 44:56. According to Neville et al. [15], CMA supplementation had a significant effect on milk production and on fat and protein content.

When buffer supplementation did not have a significant effect on any of the measured parameters, one would not expect any significant differences between groups of cows or between experimental phases. In this study, however, the differences were significant for some parameters. This was the case for differences between Groups A and B (milk protein, FT, RT, FFDM, and FFA in milk and valeric acid in rumen fluid) and between P1 and P2 (TMR consumption, FT, RT, rumen juice pH, total VFAs, and fecal pH and starch content). This is probably due to the high variability among the cows. This was reflected, for example, in the high coefficient of variation (vc), especially for daily milk yield (vc = 12.8), milk fat content (vc = 15.2), TMR consumption (vc = 13.5%), and RT per day (vc = 18.6%), but especially for FT measured by the sensor in the collar (NC) of the cow (vc = 40.8%). High coefficients of variation were found in the laboratory assessment of milk quality for fat content (vc = 16.9%), urea content (vc = 34.6%), and VFAs (vc = 36.0%). In rumen fluid, only ammonia nitrogen had a higher vc (vc = 11.7%). It is generally considered that the coefficient of variation should be less than 10%.

According to Zhang et al. [20], the reason for the higher inter-cow variability may be differences in rumination time (RT). This is because each cow may respond differently to rumen acidity. They conducted an experiment with sheep. The sheep in the SARA-sensitive group had longer RT than those in the SARA-tolerant group. Rumination was probably used as a means of mitigating ruminal acidity.

Gao and Oba [22] found in an experiment with 16 cannulated dairy cows that although DMI, MY, concentration, and volatile fatty acid profile did not differ between groups, milk urea nitrogen concentration was higher in tolerant cows compared to sensitive cows, which is probably attributable to lower fermentation of organic matter in the rumen of tolerant cows. These results suggest that there is substantial variability in SARA among lactating dairy cows fed the same high-grain TMR and that cows tolerant of the high-grain diet may be characterized by shorter chewing times and higher milk urea nitrogen concentrations. Gao and Oba [22] expected that milk fat content could be a non-invasive marker to identify tolerant and sensitive cows in the herd as cows that are tolerant to a highly fermentable diet reportedly have higher milk fat content compared to cows that are sensitive to a high cereal diet. However, milk fat content did not differ between the groups in their study. Similarly, no effect of rumen pH on milk fat was observed by other researchers, e.g., Gozho et al. [46]. This suggests that milk fat depression does not always accompany SARA.

Apparent differences between published studies are set straight by in a recent meta-analysis [13], which has drawn more objective and convincing conclusions by comparing the results. Both the meta-analysis and this study showed that buffer supplementation had no significant effect on DMI and MY. However, their findings that buffer supplementation significantly affects milk fat and lactose were not confirmed in our study.

Our findings and relevant scientific literature should be taken into account when evaluating data from sensors. Research shows great potential for smart technologies to help farmers monitor their animals’ behavior, but in real life they cannot yet be relied on 100%.

## 5. Conclusions

The inclusion of a CMA-based buffer in a well-balanced peak lactation cow diet did not have a significant impact on the performance and health parameters under study. During the whole experiment, the rumen pH and volatile fatty acids were within the normal ranges and the cows were not at risk of SARA. In spite of that, the total time of feed ingestion and rumination was increased in the cows that received the buffer during the first 3-week experimental phase. This implies that the buffer effect is delayed and persists after the withdrawal. In the group of cows that received control TMR without buffer in the first phase, RT and milk protein content increased significantly in the first week after the addition of buffer.

## Figures and Tables

**Table 1 animals-14-00897-t001:** Randomization of dairy cows (*n* = 34) into groups (2).

Index	Group A		Group B		SEM	*p*-Value
(Units)	Average	sd	Average	sd		(*n* = 34)
Number of lactations	2.41	1.12	2.35	1.06	0.26	0.876
DIM	87.4	33.4	102.2	80.2	14.9	0.485
MY (kg)	40.5	5.60	40.3	6.90	1.53	0.929
LW (kg)	689	95.6	687	63.3	19.7	0.939

DIM = days in milk; MY = daily milk yield; LW = live weight; SEM = standard error of the mean; and sd = standard deviation.

**Table 2 animals-14-00897-t002:** (**a**) Total mixed ration (TMR) composition (in kg) and bromatological analysis of components. (**b**) Chemical analysis of total mixed ration (TMR).

(**a**)							
**Component**	**TMR-C**	**TMR-E**	**DM**	**CP**	**ADF**	**NDF**	**Ash**
Maize silage	17.5	17.5	311	100	306	508	44.4
Alfalfa silage	7.0	7.0	474	248	296	390	99.3
LCWS	5.0	5.0	310	159	357	493	71.8
WBG	6.0	6.0	235	278	271	801	43.6
HMC	4.5	4.5	602	97.6	51.9	150	14.6
MGP	3.5	3.5	-	-	-	-	-
CFM-Control/CFM-CMA	8.5	8.5	-	-	-	-	-
Product Z	0.1	0.1	-	-	-	-	-
(**b**)							
Analysis index	Unit		TMR-C		TMR-E		
Dry matter	g/kg		460		460		
CP	g/kg DM		177		173		
ADF	g/kg DM		128		110		
NDF	g/kg DM		242		287		
Starch	g/kg DM		293		295		
Ca	g/kg DM		10.4		10.6		
P	g/kg DM		3.65		2.98		
Na	g/kg DM		3.50		3.60		
K	g/kg DM		11.2		10.7		
PDIA/CP			30.3		29.5		
Ca/P			2.84		3.57		
K/Na			3.15		2.97		

DM = dry matter in g/kg fresh matter; CP = crude protein in g/kg DM; ADF = acid detergent fiber in g/kg DM; NDF = neutral detergent fiber in g/kg DM; Ash = in g/kg DM; LCWS = Legume-cereal wholecrop silage 32%; HMC = High-Moisture Corn; WGB = Wet Brewers’ Grains; MGP = liquid energy feed; CFM = concentrated feed mixture; and Product Z = blend of calcium carbonate, sodium bicarbonate, sodium chloride, clinoptilolite, and magnesium oxide (pHix-up) at the lowest recommended dose 100 g/cow/day. CP = crude protein; ADF = acid detergent fiber; NDF = neutral detergent fiber; PDIA/CP = dietary protein undegraded in the rumen but truly digestible in the small intestine/CP; Ca = calcium; P = phosphorus; Na = sodium; and K = potassium.

**Table 3 animals-14-00897-t003:** (**a**) Concentrated feed composition (in %) and bromatological analysis of components. (**b**) Chemical analysis of concentrated feed.

(**a**)										
**Component**	**CFM-C**	**CFM-E**	**DM**	**CP**	**CF**	**NEL**	**Ca**	**P**	**Na**	**K**
Wheat grain	37.5	34.9	870	177	32.7	8.8	0.7	4.0	0.3	5.3
Barley grain	15.0	15.0	878	121	52.7	8.02	1.0	3.9	0.2	5.3
Soybean meal (non-GMO)	12.0	12.0	881	493	74.6	8.03	4.0	7.2	0.4	23.8
Rapeseed meal 00	24.0	24.3	890	400	135	6.96	5.6	10.1	0.4	11.9
C16–fat	3.5	3.5	-	-	-	-	-	-	-	-
MFS	8.0	8.0	-	-	-	-	25	70	120	-
Buffer CMA	0	2.3	-	-	-	-	-	-	-	-
(**b**)										
Analysis Index	Unit			CFM-C				CFM-E		
Dry matter	g/kg			860				863		
Crude protein	g/kg DM			228				224		
Crude fiber	g/kg DM			60.7				60.2		
NEL	MJ/kg DM			8.59				8.35		
Ca	g/kg DM			30.6				22.8		
P	g/kg DM			5.60				5.64		
Na	g/kg DM			7.87				7.61		
K	g/kg DM			8.70				8.79		
Ca/P				4.04				5.43		
K/Na				1.16				1.14		

CFM-C = concentrated feed control; CFM-E = CFM with the CMA buffer; DM = dry matter in g/kg fresh matter; CP = crude protein in g/kg DM; CF = crude fiber in g/kg DM; NEL = netto energy of lactation in MJ/kg DM; minerals = in g/kg DM; MFS = mineral feed supplement; CMA = blend of calcareous marine algae (*Lithotamium calcareum*), Yucca Schidigera powder, cristobalite, fulvic acid (from the drinking water industry), preservatives: calcium propionate (E282) 0.00025%, aromatic substances: blend of aromatic ingredients 0.01%, anti-caking ingredients: Silica absorbs water, (E551a) 0.35%, and carrier: wheat flour. DM = dry matter; NEL = netto energy of lactation; Ca = calcium; P = phosphorus; Na = sodium; and K = potassium.

**Table 4 animals-14-00897-t004:** Sensor values by TMR (TMR-C, TMR-E), phase (P1, P2), and group (A, B) (*n* = 34).

Parameter (Unit)	TMR		Phase		Group		SEM	*p*-Value		
	E	C	P1	P2	A	B		TMR	Phase	Group
MY (kg/cow/day)	38.8	39.2	39.6	38.4	38.9	39.1	0.49	0.518	0.072	0.703
Milk fat (%)	2.87	2.86	2.83	2.90	2.86	0.56	0.04	0.910	0.260	0.748
Milk protein (%)	2.63	2.65	2.65	2.63	2.60 ^a^	2.67 ^b^	0.01	0.431	0.443	0.002
Live weight of cows (kg)	689	689	688	690	690	689	7.8	0.980	0.831	0.930
Intake of TMR (kg/cow/day)	48.0	48.9	46.8 ^a^	50.1 ^b^	49.1	47.8	0.63	0.293	<0.001	0.148
Visits to TFTs	71.5	69.7	66.8	74.3	69.8	71.3	2.20	0.572	0.016	0.623
FT in TFTs (minutes/day)	179	172	179	172	169	181	4	0.226	0.280	0.280
FT (minutes/day) in NC	192	194	182 ^a^	203 ^b^	202 ^b^	184 ^a^	7.77	0.431	<0.001	<0.001
RT (minutes/day) in NC	460	454	492 ^b^	421 ^a^	486 ^a^	428 ^b^	7.61	0.580	<0.001	<0.001

TMR = total mixed ration; E = experimental; C = control; MY = milk yield; TFT = tensometric feed trough; NC = neck collar; FT = feeding time; RT = rumination time; and means within a row with different superscripts differ (*p* < 0.05).

**Table 5 animals-14-00897-t005:** Values from sensors by phase (P): beginning (s), middle (c), and end (e), respectively, in the first, second, and third week of each phase P1 and P2.

Parameter (Unit)	Group A						Group B						SEM	*p*-Value
	TMR-E			TMR-C			TMR-C			TMR-E				
	P1s	P1c	P1e	P2s	P2c	P2e	P1s	P1c	P1e	P2s	P2c	P2e		
MY (kg/cow/day)	40.3	38.9	38.6	40.6	39.5	39.9	38.4	38.9	38.0	38.7	37.8	38.3	1.22	0.985
Milk fat (%)	2.78	2.86	2.84	2.84	2.91	2.76	2.82	2.89	2.95	2.94	2.91	2.89	0.11	0.927
Milk protein (%)	2.60 ^a^	2.60 ^a^	2.61 ^ab^	2.67 ^ab^	2.69 ^b^	2.70 ^b^	2.63 ^ab^	2.60 ^a^	2.58 ^a^	2.69 ^b^	2.64 ^ab^	2.64 ^ab^	0.03	0.017
Live weight of cows (kg)	688	690	688	686	684	692	696	695	681	679	690	701	19.4	0.960
Intake of TMR (kg/cow/day)	42.6	47.9	50.5	41.0	48.0	50.9	51.6	52.1	49.9	46.0	52.3	48.6	1.40	0.171
Visitis to TFT	62.1	66.0	72.6	61.4	66.4	72.2	78.1	71.5	68.5	73.5	78.7	75.8	5.41	0.867
FT in TFT (minutes/day)	169	175	183	165	185	192	179	157	151	177	194	174	10	0.140
FT (minutes/day) in NC	192	179	200	174	164	185	212	226	202	192	203	184	19.3	0.748
RT (minutes/day) in NC	500 ^cd^	453 ^bcd^	447 ^bcd^	531 ^d^	497 ^cd^	526 ^d^	412 ^ab^	407 ^ab^	350 ^a^	464 ^bcd^	476 ^bcd^	419 ^abc^	18.1	<0.001

TMR = total mixed ration; E = experimental; C = control; MY = milk yield; TFT = tensometric feed trough; NC = neck collar; FT = feeding time; and RT = rumination time; means within a row with different superscripts differ (*p* < 0.05).

**Table 6 animals-14-00897-t006:** Correlation values (*p* < 0.001) for parameters measured by the sensors.

Parameter	MY	Intake of TMR	Entries into TFT	FT in TFT	FT in NC
Intake of TMR	0.451				
Visits to TFT	0.010	0.192			
FT in TFT	0.067	−0.036	0.344		
FT in NC	0.135	0.019	0.397	0.633	
RT in NC	0.181	−0.169	−0.002	0.222	0.033

TMR = total mixed ratio; MY = milk yield; TFT = tensometric feed trough; NC = neck collar; FT = feeding time; and RT = rumination time.

**Table 7 animals-14-00897-t007:** Milk quality parameters analyzed in the laboratory (*n* = 34).

		TMR		Phase		Group		SEM	*p*-Value		
Index	Units	E	C	P1	P2	A	B		TMR	Phase	Group
Fat	g/100 g	3.66	3.44	3.55	3.55	3.51	3.60	0.10	0.147	0.984	0.553
Protein	g/100 g	3.01	3.09	3.00	3.10	2.99	3.11	0.05	0.251	0.190	0.112
Lactose	g/100 g	5.04	5.06	5.07	5.02	5.02	5.07	0.03	0.586	0.207	0.165
FFDM	g/100 g	8.68	8.78	8.70	8.76	8.64 ^a^	8.82 ^b^	0.06	0.219	0.472	0.042
Casein	g/100 g	2.33	2.40	2.31	2.42	2.29	2.43	0.05	0.383	0.151	0.054
Urea	mg/100 mL	31.4	30.0	30.8	30.5	31.4	5.2	0.72	0.153	0.778	0.162
FFA	mmol/100 g	1.29	1.18	1.24	1.22	1.12 ^a^	1.34 ^b^	0.07	0.281	0.834	0.039

FFDM = fat-free dry matter, and FFA = free fatty acids; means within a row with different superscripts differ (*p* < 0.05).

**Table 8 animals-14-00897-t008:** Rumen fluid values in selected dairy cows (*n* = 12).

Index	TMR		Period		Group		SEM	*p*-Value		
(Units)	E	C	P1	P2	A	B		TMR	Period	Group
pH	6.22	6.35	6.57 ^b^	6.00 ^a^	6.29	6.29	0.10	0.382	<0.001	0.977
Total acids (mM)	133.8	130.6	122.7 ^a^	141.7 ^b^	131.6	132.7	3.27	0.502	<0.001	0.812
Lactic acid (mol%)	0.77	0.72	0.64 ^a^	0.84 ^b^	0.75	0.73	0.03	0.326	<0.001	0.747
Acetic acid (mol%)	63.3	63.7	64.1 ^b^	62.9 ^a^	63.9	63.1	0.35	0.444	0.022	0.120
Propionic acid (mol%)	19.6	19.7	19.3	20.0	19.4	19.9	0.32	0.764	0.125	0.332
Butyric acid (mol%)	16.5 ^b^	16.0 ^a^	16.0 ^a^	16.5 ^b^	16.1	16.5	0.15	0.035	0.037	0.068
Valeric acid (mol%)	0.62	0.58	0.59	0.61	0.63 ^a^	0.57 ^b^	0.02	0.139	0.475	0.039
NH_3_-N (mg N/100 g)	13.6	13.8	13.4	13.9	13.5	13.8	0.30	0.611	0.299	0.558

Means within a row with different superscripts differ at *p* < 0.05.

**Table 9 animals-14-00897-t009:** Fecal values in selected dairy cows (*n* = 12).

Index	TMR		Phase		Group		SEM	*p*-Value		
(Units)	E	C	P1	P2	A	B		TMR	Phase	Group
pH	6.58	6.59	6.67 ^b^	6.50 ^a^	6.59	6.58	0.04	0.859	0.005	0.768
Dry matter (g/kg)	144.8	148.3	149.8	143.3	150.6	142.5	4.68	0.598	0.337	0.598
Starch (g/kg DM)	33.3	34.4	35.3 ^b^	32.3 ^a^	34.2	33.4	0.63	0.246	0.003	0.246

Means within a row with different superscripts differ at *p* < 0.05.

## Data Availability

The data presented in this study are available on request from the corresponding author. The data are not publicly available due to privacy.

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
