# Peer review of "Effect of Calcareous Marine Algae Buffer on High-Producing Dairy Cows during Peak Lactation"

_animals, 2024, doi:10.3390/ani14060897_

Round 1
Reviewer 1 Report
Comments and Suggestions for Authors
The research, titled "Effect of a buffer inclusion in feed concentrate on performance and nutritional indicators in high-yielding dairy cows during peak lactation – pilot study," lacks clarity regarding its aim. Upon reading the title, one would expect a nutritional study; however, the abstract and introduction reference precision livestock farming systems, which, while they may monitor some physiological parameters of the animals, do not directly relate to the title. Consequently, I remain uncertain about the article's aim. Despite these concerns, the paper aligns with the journal's scope. Here are some suggestions for the authors to improve the article's quality:
- Rewrite the simple summary in accordance with the journal's instructions for authors (https://www.mdpi.com/journal/animals/instructions). Ensure it includes a clear statement of the problem addressed, the aims and objectives, pertinent results, conclusions from the study, and how they will be valuable to society. Additionally, it should be written for a lay audience.
- Rewrite the Abstract section, including more numerical data and significance. Additionally, clarify the aim of the research, which appears different from that indicated in lines 124-127.
- Exclude terms from the keywords already present in the title. Also, carefully review the proposed keywords, as some do not align with the article's aim.
- Rewrite the introduction, suggesting eliminating the part concerning precision livestock farming and focusing on the effect of additives on preventing SARA.
- Improve the materials and methods section, including additional information on the animals involved in the study (e.g., average age, average Body Condition Score, etc.). Also, specify how many primiparous and multiparous cows were used in the research. Some details are missing, such as the sampling frequency and how the rumen fluid was sampled. The division into phases is unclear; I suggest rewriting that part for better readability.
- Ensure Table 4 is in the results section.
- Enrich the discussion by addressing the study's limitations and practical applications.
- Avoid including citations in the conclusions section; instead, move the text to the discussion section.
Specific comment:
L 125: Provide the scientific name of the algae used.
L 168: The scientific name should be italicized
Author Response
Response to review of the paper entitled: „Effect of a buffer inclusion in feed concentrate on performance and nutritional indicators in high-yielding dairy cows during peak lactation – pilot study“.
Dear reviewer,
Thank you very much for your inspiring suggestions. We accepted all of them and tried to incorporate them into the text.
Kind Regards
Authors
Your comment 1:
The research, titled "Effect of a buffer inclusion in feed concentrate on performance and nutritional indicators in high-yielding dairy cows during peak lactation – pilot study," lacks clarity regarding its aim. Upon reading the title, one would expect a nutritional study; however, the abstract and introduction reference precision livestock farming systems, which, while they may monitor some physiological parameters of the animals, do not directly relate to the title. Consequently, I remain uncertain about the article's aim. Despite these concerns, the paper aligns with the journal's scope. Here are some suggestions for the authors to improve the article's quality:
Response 1:
Thank you for your advice. We have attempted at adjusting the text accordingly.
We thought that the aim is the same as the title of manuscript. Nevertheless, we tried to reformulate it as follows:
The aim of the study was to investigate the effect of calcareous marine algae (Lithotamium calcareum) buffer inclusion in total mixed ration (TMR) on feeding behavior and performance of high-yielding Holstein dairy cows.
The hypothesis was that when dairy cows are added to a concentrated mixture of buffer, this would affect milk production and quality, feed consumption, eating and rumination time and other parameters such as rumen acidity or faecal composition. These are methods that can be used on the farm to monitor the animals on a daily basis and to take operational action if necessary to correct the problem as soon as possible.
The correct wording should probably be that these are indicators of performance of high-yielding dairy cows during peak lactation. The word nutritional would therefore be omitted. Then the title of the study would be:
EFFECT OF CALCAREOUS MARINE ALGAE BUFFER ON HIGH-PRODUCING DAIRY COWS DURING PEAK LACTATION
You may also have found the second objective confusing: 'variability'. We are aware of the mistake. It is not about the goal, but about discussing why the result did not meet expectations. Yes, the variability between dairy cows was probably the main reason that the addition of buffer did not make a significant difference. So we are moving the variability to the discussion section.
Evaluation of the results of the daily monitoring measurements of these data showed that the issue is much more complex than it first appeared. It showed that there were large differences between individual dairy cows, both in terms of daily changes and response by indicator. The fact that the TMR was optimised and thus the cows did not get into an acidic state probably played a significant role. However, this is very difficult to estimate in advance. Not to mention that according to the Animal Care and Use Committee, the experiment cannot be formulated in such a way that the dairy cows were deliberately fed in such a way that SARA became more pronounced.
Your comment 2:
- Rewrite the simple summary in accordance with the journal's instructions for authors (https://www.mdpi.com/journal/animals/instructions). Ensure it includes a clear statement of the problem addressed, the aims and objectives, pertinent results, conclusions from the study, and how they will be valuable to society. Additionally, it should be written for a lay audience.
Response 2:
We have rewritten the whole simple summary.
Your comment 3:
- Rewrite the Abstract section, including more numerical data and significance. Additionally, clarify the aim of the research, which appears different from that indicated in lines 124-127.
Response 3:
We have rewritten the whole Abstract.
Your comment 4:
Exclude terms from the keywords already present in the title. Also, carefully review the proposed keywords, as some do not align with the article's aim.
Response 4:
We corrected keywords.
Your comment 5:
- Rewrite the introduction, suggesting eliminating the part concerning precision livestock farming and focusing on the effect of additives on preventing SARA.
Response 5:
We have deleted the some text from Introduction chapter. We have included only some references to the literature and used some information in the discussion.
Your comment 6:
- Improve the materials and methods section, including additional information on the animals involved in the study (e.g., average age, average Body Condition Score, etc.). Also, specify how many primiparous and multiparous cows were used in the research. Some details are missing, such as the sampling frequency and how the rumen fluid was sampled. The division into phases is unclear; I suggest rewriting that part for better readability.
Response 6:
The average age is determined by lactation order, which is shown in Table 1. There were 4 heifers in each group, which is shown in L211. Dairy cow weight was used in place of body condition score and was measured twice for each cow each day as the cows passed over an automatic scale on their way to the parlor. This is a more accurate figure than the Body Condition Score, which we would have had to estimate, and we certainly would not have been able to estimate this figure every day for all dairy cows.
Samples of feed, milk, rumen fluid and faeces were collected for chemical analysis 3 times, the first time before the start of the experiment in the preparatory period and subsequently in each period. This information is in L267 to L269. Milk samples were collected in the milking parlor for spectroscopic analysis twice daily, always at the time of milking of each cow. This information is in L297.
Rumen fluids were collected from dairy cows using a rumen probe. This information is in L303-313.
For readability, we report that there was a preparatory period before the experiment, followed by period 1 and then period 2. At the end of P1, the TMR of group A and group B changed from experimental to control and control to experimental, respectively.
This is stated in other words in the annotation and in more detail in the methodology.
The design of this experiment was comparable to the design of experiment Loucka at al. (2023), which is reported in L174. Because the ORIGINALITY REPORT indicated that the similarity index was 9% with cjas.agriculturejournals.com, this passage was significantly abbreviated by reference to the methods reported in the Loučka at al. (2023) study,
Your comment 7:
Ensure Table 4 is in the results section.
Response 7:
We moved Table 4 to the |Results section.
Your comment 8:
Enrich the discussion by addressing the study's limitations and practical applications.
Response 8:
We enriched the discussion by addressing the limitations of the study and practical applications as follows:
Whether the addition of buffer to TMR will be effective depends on other factors as well, especially the individuality of the animals and their response to a given stimulus at a given time. The outcome may also be influenced by the continuous measurement of performance and health indicators by methods that can be used to promptly take action on the farm to correct any identified problems as soon as possible. An important reason that the addition of buffer did not affect the measured parameters could be that the dairy cows were given a TMR that did not induce SARA. (L475 to L480).
The study is a pilot study. For the next experiment, it will be necessary to change the conditions, e.g. include the buffer in a different TMR.
Your comment 9:
Avoid including citations in the conclusions section; instead, move the text to the discussion section.
Response 9:
The whole paragraph has been moved to the discussion as per your recommendation.
Your comment 10:
Specific comment:
L 125: Provide the scientific name of the algae used.
Response 10:
It was done.
Your comment 11:
L 168: The scientific name should be italicized
Response 11:
This was done throughout the document.
Reviewer 2 Report
Comments and Suggestions for Authors
Manuscript number: animals-2880812
Title: Effect of a buffer inclusion in feed concentrate on performance and nutritional indicators in high-yielding dairy cows during peak lactation – pilot study.
Even though this manuscript is interesting, a several concern needs to be addressed before further consideration. Please see the specific comment below.
Introduction
Line 47: Discuss all the factors contributing to SARA's rise or decline.
Line 94: Introduce further details in the area dedicated to the Neck collar, including the operational principles of the mechanism and the existence of any analogous technologies, as it has been previously used.
Materials and Methods
Line 126-127: Additional details are provided in the section elucidating the mean milk content derived from Table 1.
Line 140: The PLF method: What is the nature or identity of the subject in question? The principles refer to fundamental concepts or guidelines that serve as the foundation for a system or process. Mechanisms, on the other hand, are the specific methods or processes used to achieve a certain outcome or goal. Please elaborate on the topic.
Line 144-146: Elucidate the fundamental concepts and processes used by the machine utilized for the acquisition and examination of samples.
Line 148: Which standards or criteria are employed to ascertain the mean milk content of the cattle utilized in the experiment?
Lines 150-151: Is the quantity of calves assigned to each treatment sufficient? Separate explanations should be provided for the number of calves utilized by each treatment.
Table
Please ensure that the decimal position remains consistent across all values in the table.
P or P-value? Please check.
P-value should be in italic style.
SD: Capital letters ought to be utilized.
Tables 2 and 3
Specify the quantities of Ca, P, Na, and K utilized in the calculations to derive various table values.
Line 221-222: Specify the values in the section devoted to (#934.01) and (#920.40). From where do they originate within the table? To what extent are the units?
Discussion
There is insufficient elucidation regarding the values appended to Table 6
Comments on the Quality of English LanguageMinor editing of English language required
Author Response
Response to review of the paper entitled: „Effect of a buffer inclusion in feed concentrate on performance and nutritional indicators in high-yielding dairy cows during peak lactation – pilot study“.
Dear reviewer,
Thank you very much for your inspiring suggestions. We accepted all of them and tried to incorporate them into the text.
Kind Regards
Authors
Your comment 1:
Even though this manuscript is interesting, a several concern needs to be addressed before further consideration. Please see the specific comment below.
Response 1:
Thank you for your comment. We use your advice.
Your comment 2:
Introduction
Line 47: Discuss all the factors contributing to SARA's rise or decline.
Response 2:
According to a meta-analysis [9], there are many factors that can influence SARA. The main ones are cow factor and dietary error feeding factor. No single dietary factor solely drives the physiochemistry of the rumen, but is a combination of several factors. L90 to L92
Your comment 3:
Line 94: Introduce further details in the area dedicated to the Neck collar, including the operational principles of the mechanism and the existence of any analogous technologies, as it has been previously used.
Response 3:
An indirect method for diagnosing SARA is to measure chewing movements using an accelerometer placed in the cervical collar. In addition to monitoring chewing movements, the most practical indirect indicators for the diagnosis of SARA include the appearance of faeces and the fat/protein ratio of milk. For more details, see Mattachini et al. (2016). L118 to L121.
Mattachini, G., Riva, E., Perazzolo, F., Naldi, E., & Provolo, G. (2016). Monitoring feeding behaviour of dairy cows using accelerometers. Journal of Agricultural Engineering, 47(1), 54-58.
Your comment 4:
Materials and Methods
Line 126-127: Additional details are provided in the section elucidating the mean milk content derived from Table 1.
Response 4:
MY averaged more than 40 litres at the start of the trial. At the beginning of the trial, the average fat content of milk for both groups was 2.6% and protein content was 2.8%. L213 to L217
Your comment 5:
Line 140: The PLF method: What is the nature or identity of the subject in question? The principles refer to fundamental concepts or guidelines that serve as the foundation for a system or process. Mechanisms, on the other hand, are the specific methods or processes used to achieve a certain outcome or goal. Please elaborate on the topic.
Response 5:
This paragraph has been removed on the recommendation of another opponent.
Your comment 6:
Line 144-146: Elucidate the fundamental concepts and processes used by the machine utilized for the acquisition and examination of samples.
Response 6:
This paragraph has also been removed on the recommendation of another opponent. Milk sampling is done automatically by the Afimilk system, chemical analysis of samples is done automatically by the Afilab module.
Your comment 7:
Line 148: Which standards or criteria are employed to ascertain the mean milk content of the cattle utilized in the experiment?
Response 7:
Milk quality was analyzed in an accredited (ČSN EN ISO/IEC 17025:2005, 2018) laboratory (MILCOM a.s. Prague, CR). Infrared spectrophotometry (ČSN 57 0530, 2010; ČSN 57 0536, 1999) was used to analyze fat, protein, lactose and non-fat dry matter of milk; casein, urea and free fatty acids (FFA) by the indirect MIR-FT method (Hanuš et al. 2008). L299 to L302.
Your comment 8:
Lines 150-151: Is the quantity of calves assigned to each treatment sufficient? Separate explanations should be provided for the number of calves utilized by each treatment.
Response 8:
All automatic tasks were measured daily for all dairy cows. There were only as many dairy cows as the capacity of the barn, which is 34. Manual measurements, i.e. check sampling of rumen fluid, milk for chemical analysis and faeces composition, were carried out on 12 cows for operational reasons in the second and subsequent lactations.
Your comment 9:
Table
Please ensure that the decimal position remains consistent across all values in the table.
P or P-value? Please check.
P-value should be in italic style.
SD: Capital letters ought to be utilized.
Response 9:
It was done. SD was deleted in all tables on the recommendation of another reviewer.
Your comment 10:
Tables 2 and 3
Specify the quantities of Ca, P, Na, and K utilized in the calculations to derive various table values.
Response 10:
The values of Ca, P, Na, and K were added to both tables.
Your comment 11:
Line 221-222: Specify the values in the section devoted to (#934.01) and (#920.40). From where do they originate within the table? To what extent are the units?
Response 11:
The numbers in brackets have been removed from the document. L314 to 315
Your comment 12:
Discussion
There is insufficient elucidation regarding the values appended to Table 6
Response 12:
For a discussion of this table, see L 518 to 525
At the same time we have made the table clearer.
Your comment 13:
Comments on the Quality of English Language
Minor editing of English language required
Response 13:
English has been edited, but not by a native speaker.
Reviewer 3 Report
Comments and Suggestions for Authors
Although results have been published on buffers in general and on calcareous seaweed specifically, the work has enough novelty to make it worth publishing. Overall, the work was well presented, requiring only a few adjustments for better reader understanding. The study also addresses a relevant topic, requiring more research involving animals in other stages of lactation, in addition to observing a longer lactation period. Still regarding the interpretation of the results, some comments can be found below.
Key-work
Line 42 - very generalist keywords
Introduction
The objective must be contained in the introduction.
Line 66 - What are these sensors? Please explain further
The introduction is too long.
Materials and Methods
Approval from the Animal Care and Use Committee is lacking.
Line 174 - Table 2, rearrange the table by placing the analysis below the composition to facilitate understanding.
Line 196 - Table 3, rearrange the table, same as Table 2 (line 174)
Results
Did groups A and B participate in both phases (P1 and P2)? Please explain in more detail how this distribution was conducted.
Line 252 - Table 5 shows that group A only had a relationship with phase 1 and group B with phase 2. Please explain in more detail how this analysis was conducted.
Conclusions
The conclusion seems more like a discussion, where the findings corroborate existing studies. I suggest a greater focus on addressing your hypothesis.
Author Response
Response to review of the paper entitled: „Effect of a buffer inclusion in feed concentrate on performance and nutritional indicators in high-yielding dairy cows during peak lactation – pilot study“.
Dear reviewer,
Thank you very much for your inspiring suggestions. We accepted all of them and tried to incorporate them into the text.
Kind Regards
Authors
Although results have been published on buffers in general and on calcareous seaweed specifically, the work has enough novelty to make it worth publishing. Overall, the work was well presented, requiring only a few adjustments for better reader understanding. The study also addresses a relevant topic, requiring more research involving animals in other stages of lactation, in addition to observing a longer lactation period. Still regarding the interpretation of the results, some comments can be found below.
We chose the peak of lactation because SARA is most common during this period. Later on, SARA usually manifests itself as a result of a gross error.
Your comment 1:
Key-work
Line 42 - very generalist keywords
Response 1:
We have better specified the keywords.
Your comment 2:
Introduction
The objective must be contained in the introduction.
Line 66 - What are these sensors? Please explain further
Response 2:
The aim has been added to the introduction. L122 to L124
The sentence regarding sensors and PLF was removed based on the recommendation of another opponent.
Your comment 3:
The introduction is too long.
Response 3:
The introduction has been significantly shortened.
Your comment 4:
Materials and Methods
Approval from the Animal Care and Use Committee is lacking.
Response 4:
WE added: The experimental protocol was approved by the institutional Animal Care and Use Committee, Institute of Animal Science, Prague, Czech Republic (Act No. 359/2012 Coll.). L204 to 205
Your comment 5:
Line 174 - Table 2, rearrange the table by placing the analysis below the composition to facilitate understanding.
Response 5:
It was done.
Your comment 6:
Line 196 - Table 3, rearrange the table, same as Table 2 (line 174)
Response 6:
It was done.
Your comment 7:
Results
Did groups A and B participate in both phases (P1 and P2)? Please explain in more detail how this distribution was conducted.
Response 7:
Yes, groups A and B participate in both phases (P1 and P2). There was an adaptation period before the start of the experiment, followed by phase 1 and then phase 2. At the end of P1, the TMR of group A and group B changed from experimental to control and control to experimental, respectively. Described at L227 to 233
Your comment 8:
Line 252 - Table 5 shows that group A only had a relationship with phase 1 and group B with phase 2. Please explain in more detail how this analysis was conducted.
Response 8:
Explained below the table.
Your comment 9:
Conclusions
The conclusion seems more like a discussion, where the findings corroborate existing studies. I suggest a greater focus on addressing your hypothesis.
Response 9:
This section has been moved to the discussion section.
As per your recommendation, we have focused on justifying why the addition of the buffer did not have a significant effect.
L644 to 652
Reviewer 4 Report
Comments and Suggestions for Authors
I find the manuscript interesting, however it requires some aspects to be addressed to be published in the journal.

Author Response
Response to review of the paper entitled: „Effect of a buffer inclusion in feed concentrate on performance and nutritional indicators in high-yielding dairy cows during peak lactation – pilot study“.
Dear reviewer,
Thank you very much for your inspiring suggestions. We accepted all of them and tried to incorporate them into the text.
Kind Regards
Authors
Your comment 1:
What is the reason why a complete bromatological analysis is not carried out on each of the ingredients? Only one parameter of each of the ingredients is presented, it is important to know the complete bromatological characteristics.
Response 1:
We have divided the table into 2 tables. We have added bromatological analysis to table 2a, containing the main components DM, CP, ADF, NDF and Ash. To table 2b we have added the ADF values and the values of the minerals Ca, P, Na, K. This is what Reviewer 2 asked for.
Your comment 2:
Fiber plays an important role as a pH regulator in the rumen, it seems to me that the amount of neutral detergent fiber and acid detergent fiber in each of the ingredients should be placed.
Response 2:
We have divided the table into 2 tables. We have added bromatological analysis to table 3a, containing the main components DM, CP, CF and minerals Ca, P, Na, K. We have added the values of minerals Ca, P, Na, K to both tables 3a nad 3b. This is what Reviewer 2 required. In the table 3b, fiber is represented as CF (crude fiber). ADF and values are given in the previous Table 2b, where the total mixed ration (TMR) is specified.
Your comment 3:
It seems to me that the statistical analysis can be a 2x2x2 factorial design, which would allow us to see the interactions between the factors and how they influence the results.
Response 3:
We replaced the multifactorial design with a 2x2x2 factorial design. L322
Your comment 4:
It is suggested to eliminate the standard deviation, because we have the standard error of the mean, which is finally the estimator of the standard deviation.
Response 4:
We eliminated it in all tables.
Your comment 5:
Eliminate 2 sentences.
Response 5:
We eliminated it.
Your comment 6:
This is a discussion, not a conclusion.
Response 6:
We moved the text to the Discussion chapter.
Round 2
Reviewer 1 Report
Comments and Suggestions for Authors
The manuscript has been improved by addressing the reviewers' concerns. However, there are still some points that need revision:
- I advise against using abbreviations in the simple summary, as requested in the author guidelines.
LL 56-58: Please provide a citation. For example, I suggest citing the paper: 1828051X.2021.1990804.
LL 76-80: I don't understand the relevance of PLF to the article's topic. In my opinion, this part should be removed.
LL 86-87: This sentence doesn't seem relevant. I recommend removing it.
L153: Please include the units of measurement for the bromatological analyses.
L160: "not analyzed"?
L 188: "not analyzed"?
L 213-215: Please review the sentence and rewrite it for clarity.
L 317: The abbreviation has already been explained.
L 365: Remove "2020."
L 379: Remove "2018."
Comments on the Quality of English LanguageI suggest that the authors carefully review the text to eliminate spelling errors.
Author Response
Response to review of the paper entitled: „Effect of a buffer inclusion in feed concentrate on performance and nutritional indicators in high-yielding dairy cows during peak lactation – pilot study“.
Dear reviewer,
Thank you very much for your inspiring suggestions. We tried to incorporate them into the text.
Kind Regards
Authors
Your comment 1:
I advise against using abbreviations in the simple summary, as requested in the author guidelines.
Response 1:
OK, we removed (SARA) in L15; (CMA) in L17; (TMR) in L18; (feeding time, FT; rumination time, RT) in L25;
Instead of CMA in L21, we put buffer; Instead of TMR-E and TMR-C in L22, we put experimental and control total mixed ration; Instead of TMR in L30, we put buffer Instead of SARA in L30, we put acidosis.
Your comment 2:
LL 56-58: Please provide a citation. For example, I suggest citing the paper: 1828051X.2021.1990804.
Response 2:
We cited suggested paper. L61
Your comment 3:
LL 76-80: I don't understand the relevance of PLF to the article's topic. In my opinion, this part should be removed.
Response 3:
We removed it.
Your comment 4:
LL 86-87: This sentence doesn't seem relevant. I recommend removing it.
Response 4:
We removed it.
Your comment 5:
L153: Please include the units of measurement for the bromatological analyses.
Response 5:
Units are placed in footnotes. We are convinced that presentation in table title would leads to too long title. Inside the table, cells would be shifted.
Your comment 6:
L160: "not analyzed"?
Response 6:
Parameters described as "na" was not analysed for some ration ingredients, so it is reason why it is not presented. These ingredients do not content these nutrients. However it is true that it is misleading so we deleted “na” and gave there “-“.
Your comment 7:
L 188: "not analyzed"?
Response 7:
Parameters described as "na" was not analysed for some ration ingredients, so it is reason why it is not presented. These ingredients do not content these nutrients. However it is true that it is misleading so we deleted “na” and gave there “-“.
Your comment 8:
L 213-215: Please review the sentence and rewrite it for clarity.
Response 8:
We rewrote these sentences. L212-215
Your comment 9:
L 317: The abbreviation has already been explained.
Response 9:
We deleted it.
Your comment 10:
L 365: Remove "2020."
Response 10:
We removed it.
Your comment 11:
L 379: Remove "2018."
Response 11:
We removed it.
Your comment 12:
Comments on the Quality of English Language
I suggest that the authors carefully review the text to eliminate spelling errors.
Response 12:
We did some text corrections in the manuscript.
Reviewer 2 Report
Comments and Suggestions for Authors
Thanks to the authors for incorporating comments into the manuscript. There is a minor comment remaining and please see in dpf file.

Author Response
Response to review of the paper entitled: „Effect of a buffer inclusion in feed concentrate on performance and nutritional indicators in high-yielding dairy cows during peak lactation – pilot study“.
Dear reviewer,
Thank you very much for your inspiring suggestions. We tried to incorporate them into the text.
Kind Regards
Authors
Your comment 1:
Please add a sentence of conclusion at the end.
Response 1:
We added it. L46-L49
Your comment 2:
Please check!
Response 2:
We deleted it.
Your comment 3:
Add 1-2 citations!
Response 3:
We added 2 citations. L58 and L61
Your comment 4:
Only one study? not many!..please clarify.
Response 4:
We corrected it. L73
Your comment 5:
Add hypothesis before objective.
Response 5:
Hypothesis was added. L86-L88
Your comment 6:
check
Response 6:
It was deleted.